# Effect of different designs of interspinous process devices on the instrumented and adjacent levels after double-level lumbar decompression surgery: A finite element analysis

**Hao-Ju Lo** [1,2], **Hung-Ming Chen** [3], **Yi-Jie Kuo** [4,5], **Sai-Wei Yang** [1]*

1 Department of Biomedical Engineering, National Yang-Ming University, Taipei, Taiwan, 2 Department of Orthopedic Surgery, Dali Branch, Jen-Ai Hospital, Taichung, Taiwan, 3 Department of Orthopedic Surgery, Ren-Ai Branch of Taipei City Hospital, Taipei, Taiwan, 4 Department of Orthopedic Surgery, Wan Fang Hospital, Taipei Medical University, Taipei, Taiwan, 5 Department of Orthopedic Surgery, School of Medicine, College of Medicine, Taipei Medical University, Taipei, Taiwan

* swyang@ym.edu.tw

**Data Availability Statement:** All relevant data are within the paper.

## Abstract

Recently, various designs and material manufactured interspinous process devices (IPDs) are on the market in managing symptomatic lumbar spinal stenosis (LSS). However, atraumatic fracture of the intervening spinous process has been reported in patients, particularly, double or multiple level lumbar decompression surgery with IPDs. This study aimed to biomechanically investigate the effects of few commercial IPDs, namely DIAM™, Coflex™, and M-PEEK, which were implanted into the L2-3, L3-4 double-level lumbar spinal processes. A validated finite element model of musculoskeletal intact lumbar spinal column was modified to accommodate the numerical analysis of different implants. The range of motion (ROM) between each vertebra, stiffness of the implanted level, intra stress on the intervertebral discs and facet joints, and the contact forces on spinous processes were compared. Among the three implants, the Coflex system showed the largest ROM restriction in extension and caused the highest stress over the disc annulus at the adjacent levels, as well as the sandwich phenomenon on the spinous process at the instrumented levels. Further, the DIAM device provided a superior loading-sharing between the two bridge supports, and the M-PEEK system offered a superior load-sharing from the superior spinous process to the lower pedicle screw. The limited motion at the instrumented segments were compensated by the upper and lower adjacent functional units, however, this increasing ROM and stress would accelerate the degeneration of un-instrumented segments.

## Introduction

Lumbar spinal stenosis (LSS) is defined as a narrowing of the spinal canal and can cause considerable pain or numbness in the legs. The main causes of spinal stenosis are bulging of the

**Funding:** This study was funded by the Ministry of Science and Technology, ROC. (106-2221-E-010 -007 -MY2, 106-2221-E-075 -002.) The funding sources played no role in the design, implementation, data analysis, interpretation, or reporting of the study.

**Competing interests:** The first author, corresponding author, and the developer (Dr. Hsin-Chang Chen) of M-PEEK device were once colleagues in Department of Biomedical Engineering, National Yang-Ming University, Taipei, Taiwan. But there are no other relationships or activities that could appear to have influenced the submitted work.

intervertebral disc, hypertrophic facet capsular ligament, or the ligamentum flavum, which narrows the spinal cord or root. Surgical treatment of LSS typically involves resecting osteophytes and expanding the space between vertebrae, or alternatively fusing vertebrae in the region of the stenosis. The most common surgical treatment is single to multi-level decompressive laminectomy [1]. The dorsal decompression procedure provides instant chronic pain relief and allows the patient to rapidly resume everyday activities [2].

The lumbar interspinous process device (IPD) is based on this dorsal decompression principle and is placed at a symptomatic level to limit the extension of the lumbar spine and maintain a relatively flexible mobility at the intradiscal level to achieve the goal of symptom relief. In case of the patient's symptoms being not alleviated through a flexed posture, the traditional decompression surgery must be performed. Therefore IPDs are designed as a spacer to offload facet joints and release the entrapped spinal root nerves and as the intralaminar stabilizer [3–7].

The device materials vary from metals such as titanium (such as X-Stop, Coflex) to rigid polymers like polyetheretherketone (PEEK) (M-PEEK), or other elastomer compounds (DIAM, Silicone coated with Dacron). The fixation method for the device to the vertebral spinous processes may also be categorized as either static spacer, such as X-Stop (Medtronic, USA) and Wallis (Abbott Spine, France); or flexible implant such as Coflex (RTI Surgical, USA), DIAM (Medtronic Sofamor Danek, USA), and M-PEEK (a PPEK pedicle screw-based M shape interspinous spacer developed by Chen et al. [15].

The IPD is commonly implanted in single or double-level lumbar stenosis from L1-L5 in skeletally mature patients, and possible application in across multiple levels. Fracture of the spinous processes is considerable complication as placement of the device changes the mechanism of stress on the spinous process from a tension-bearing role to a compression-loaded structure during the trunk flexion, this is true particularly for single-level patients, but the risk of this complication in double-level cases has yet to be fully evaluated. Whether instrumented IPDs offer superior outcomes to non-instrumented bony decompression techniques is still controversial [8–10]. However, the advantage of ease of use and relatively short surgery time, IPD surgery is still accepted by the most of the patients. Besides the instrumented spinous fracture, a recognized complication termed the "sandwich phenomenon" also contribute to cracking of the intervening spinous process in patients with adjacent, double-level IPDs as reported with the use of two-level X-Stop [11]. In a multicenter study by Gazzeri et al. [12], the average postoperative spinous process fracture rate was about 2.05% with various IPDs implanted. Nevertheless, the patients treated with titanium X-Stop alone revealed at highest risk of fracture with an incidence rate of 3.79%. In this reason this product was off the market in 2015 [13, 14].

Therefore, the goal of this work was to look into alterations in the biomechanical characteristics of the lumbar spine after implanting three different designed IPDs across two adjacent levels and the efficacy of stress relief as well as the ROM. We hypothesized that IPDs which leads to less limiting the extension range of motion (ROM) at instrumented levels decrease the "sandwich phenomenon", and the higher limiting extension the higher stress over the disc annulus at the adjacent levels.

## Materials and methods

### FEMs of the lumbar spine and implant

The finite element software ANSYS 16.0 (ANSYS Inc., Canonsburg, PA) was used to create a 3-dimensional FE model of the 5-level intact ligamentous lumbar spine (Fig 1A). The 3D lumbar spinal column bony structure was obtained from CT images of a healthy male and then

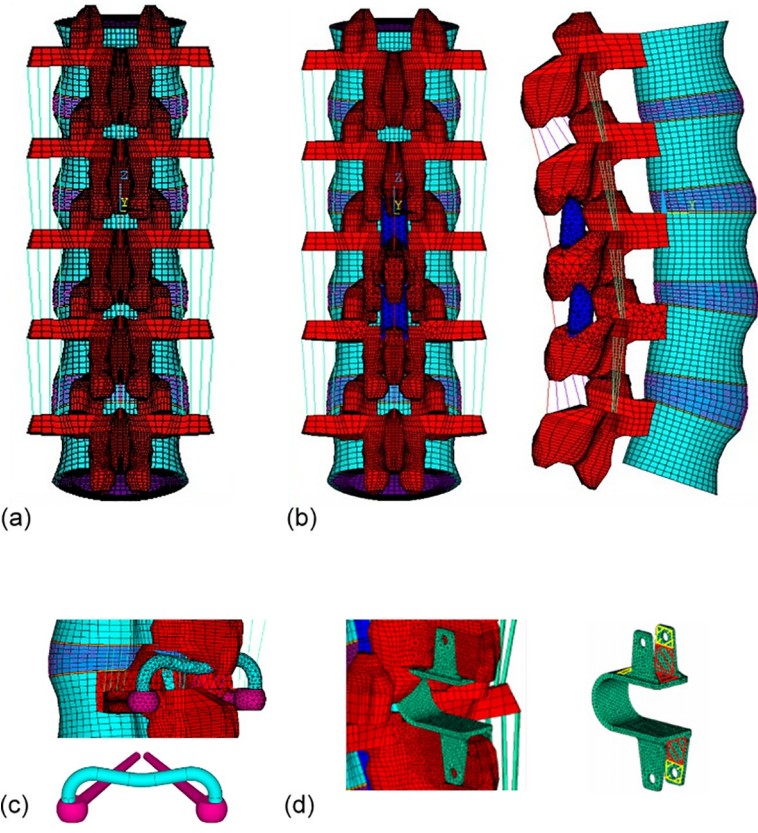

**Fig 1.** Finite element model of the (a) intact lumbar spine implanted with two interspinous spacers of (b) DIAM, (c) pedicle screw-based M-rod (M-PEEK) system and (d) Coflex at L2/L3 and L3/L4 segments.

reconstructed to the nodal points of computer FEM geometry. In order to simplify the model, a specialized command in ANSYS was used to rotate, translate and scale the L3 vertebra to reproduce the L1, L2, L4, and L5 bony structure. These vertebrae were then aligned into a lordotic lumbar spine column according to the upright X-ray images of the same person (Fig 1B). The intact model was validated by applying a pure moment of 7.5 Nm in flexion, extension, torsion, and lateral bending, respectively; all degrees of freedom at the inferior surfaces of the L5 vertebra were constrained. The ROM of each vertebra and the facet joint forces at all segments was computed. Subsequently, 1000 N axial compressive force was acting on the top of L1 vertebra and the intradiscal pressures were calculated in comparing to published literatures [15–20]. The material properties are listed in Table 1 [16–18].

The intact spinal model was then implanted with a DIAM^TM, Coflex^TM and M-PEEK in the interspinous space at L2-3 and L3-4 (Fig 1B–1D) as three simulation models, respectively. The M-PEEK system has a novel bilateral pedicle screws with diameter of 6.0 mm and a length of 45 mm into the laminar as the base pole. An "M" geometry rod with 5.5 mm diameter is attached to the base poles and the concave part is placed underneath the spinous process. The three FE implanted models were constructed according to the real implant dimension as well as material properties posted, and were validated in previous studies [15, 19, 20]. According to the surgical procedures, the Coflex and DIAM interspinous spacers were inserted between adjacent spinous processes of the lumbar spine to resist hyperextension of the lumbar spine. In the Coflex spinal model, spinal instability was simulated by cutting the ligamentum flavum, supraspinous ligament, interspinous ligament, and the facet capsules, and by removing 50% of

**Table 1. The material properties of intact spine, Coflex, M-rod and DIAM model components.**

| Material | Young's modulus (MPa) | Poisson's ratio |
|---|---|---|
| Cortical bone | 12,000 | 0.2 |
| Cancellous bone | 300/100 | 0.2 |
| Annulus fibrous | Mooney-Rivlin | NA |
| | c1 = 0.18, c2 = 0.045 | |
| Nucleus pulposus | Mooney-Rivlin | NA |
| | c1 = 0.12, c2 = 0.03 | |
| Coflex (Titanium alloy) | 113,000 | 0.3 |
| Pedicle screw (Titanium alloy) | 113,000 | 0.3 |
| M-shaped device (PEEK) | 4,000 | 0.3 |
| DIAM (compression) | 20 | 0.45 |
| DIAM (extension) | 5,000 | 0.3 |
| Ligament | | |
| ALL | 7.8 | NA |
| PLL | 10 | NA |
| TL | 10 | NA |
| LF | 15 | NA |
| ISL | 10 | NA |
| SSL | 8 | NA |
| CL | 7.5 | NA |

C1, C2 indicated two parameters of Mooney-Rivlin hyperelastic formation; NA = not applicable; ALL, anterior longitudinal ligament; CL, capsular ligament; ISL, interspinous ligament; LF, ligamentum flavum; PLL, posterior longitudinal ligament; SSL, supraspinous ligament; TL, transverse ligament.

the bilateral inferior bony facet at the L2, L3 and L4 segments. This process was repeated for the DIAM and M-PEEK models, except the supraspinous and interspinous ligaments were preserved in both cases (Fig 1B and 1C). The surface between the spinous process and the interspinous spacer was assigned as the contact surface. The heights of the interspinous space at L2-3 and L3-4 in all instrumented models were identical. Table 1 details the material properties of all implant components.

## Boundary and loading condition

The hybrid testing protocol was carried out to evaluate the effect of implanting the interspinous spacers on the interspinous and adjacent levels [21]. The intact model was subjected to a 150 N compressive follower load combined with a 9.9 Nm moment under physiological motions. The follower load was simulated by taking a two-node link element attached near the geometric center of each vertebra and maintained at a tangent to the spinal curvature. This setup was intended to mimic physiological compressive loading and persist the lumbar lordotic angle [17, 19, 22]. The loading path of the follower load was able to constrain the ROM of each segment within 0.6 degrees for all FE models.

The instrumented spinal models were loaded with the same follower load as the intact model, but the moment was incrementally increased until the total ROM resembled to that of the intact model. The resulting deviations in ROM among the four FE models were controlled to be within 0.6 degrees (Table 2). All FE models were constrained on the bottom of the fifth vertebra. Each model was compared to the intact model in terms of ROM, peak intradiscal stress and facet joint contact force. In addition, the contact forces between the devices and spinous processes and the maximum Von-Mises stress on the spinous processes were analyzed.

**Table 2. The ROMs of different implanted models in extension, flexion, axial rotation and lateral bending.**

| Model | ROMs (degree) | | | | Total lumbar ROMs (degrees) |
|---|---|---|---|---|---|
| | L1/L2 | L2/L3 | L3/L4 | L4/L5 | |
| Extension | | | | | |
| Intact | 3.05 | 2.62 | 2.56 | 2.57 | 10.8 |
| Coflex | 4.61 | 0.73 | 0.54 | 4.86 | 10.73 |
| DIAM | 3.42 | 2.17 | 2.07 | 3.23 | 10.89 |
| M-PEEK | 4.11 | 1.25 | 1.13 | 4.34 | 10.83 |
| Flexion | | | | | |
| Intact | 4.77 | 4.74 | 4.62 | 6.05 | 20.18 |
| Coflex | 4.85 | 4.64 | 4.54 | 6.14 | 20.17 |
| DIAM | 5.29 | 4.18 | 3.89 | 6.68 | 20.04 |
| M-PEEK | 4.80 | 4.74 | 4.62 | 6.03 | 20.19 |
| Axial rotation | | | | | |
| Intact | 2.01 | 2.3 | 2.68 | 3.75 | 10.74 |
| Coflex | 2.11 | 2.17 | 2.71 | 3.79 | 10.78 |
| DIAM | 2.06 | 2.3 | 2.58 | 3.84 | 10.78 |
| M-PEEK | 2.05 | 2.3 | 2.61 | 3.81 | 10.77 |
| Lateral bending | | | | | |
| Intact | 5.47 | 5.01 | 4.7 | 4.48 | 19.66 |
| Coflex | 5.71 | 4.12 | 3.81 | 6.02 | 19.66 |
| DIAM | 5.47 | 4.98 | 4.68 | 4.49 | 19.62 |
| M-PEEK | 5.47 | 5.01 | 4.7 | 4.48 | 19.66 |

## Results

### ROM at instrumented and adjacent level

In comparison to the intact model, the ROM of extension at instrumented L2-3 and L3-4 decreased in all implanted models. Among the results, the Coflex model had the most significant reduction of 72% and 79%, respectively, followed by the M-PEEK with 52% and 56% reduction, respectively. However, these limiting movement was compensated by the adjacent un-instrumented functional units, in which, the Coflex showed +51% increase at L1-2 and +89% at the L4-5; the M-PEEK showed +35%, and +69%, respectively. In flexion simulation, all three models had ROM similar to the intact one, except the DIAM system showed 12% and 16% decrease at L2-3, and L3-4 but compensated by +11%, and +10% at L1-2, and L4-5, respectively (Fig 2 and Table 2).

Compared to the intact model in axial rotation, the ROM of instrumented and adjacent levels was very similar in the DIAM and M-PEEK models. In lateral bending, the ROM differences in all instrumented models were less than 1% for each segment as compared to the intact model; for the Coflex model, the ROM was more variation as compared to the intact model, as depicted in Fig 2 and Table 2.

### Disc stress at instrumented and adjacent levels

Fig 3 presents the maximum disc stress at the instrumented and adjacent levels of the Coflex, DIAM and M-PEEK models. Compared to the intact model in extension, the peak stress at L2-3 and L3-4 instrumented levels decreased more than 40% in the Coflex and M-PEEK model. The peak stress slightly increased in adjacent levels at L1-2 in all instrumented models; it was increased more than L1-2 in L4-5 of the instrumented models. Likened to the intact model in flexion, the disc stress differences between all models was less than 1% for each segment except for the DIAM model, as expressed in Fig 3.

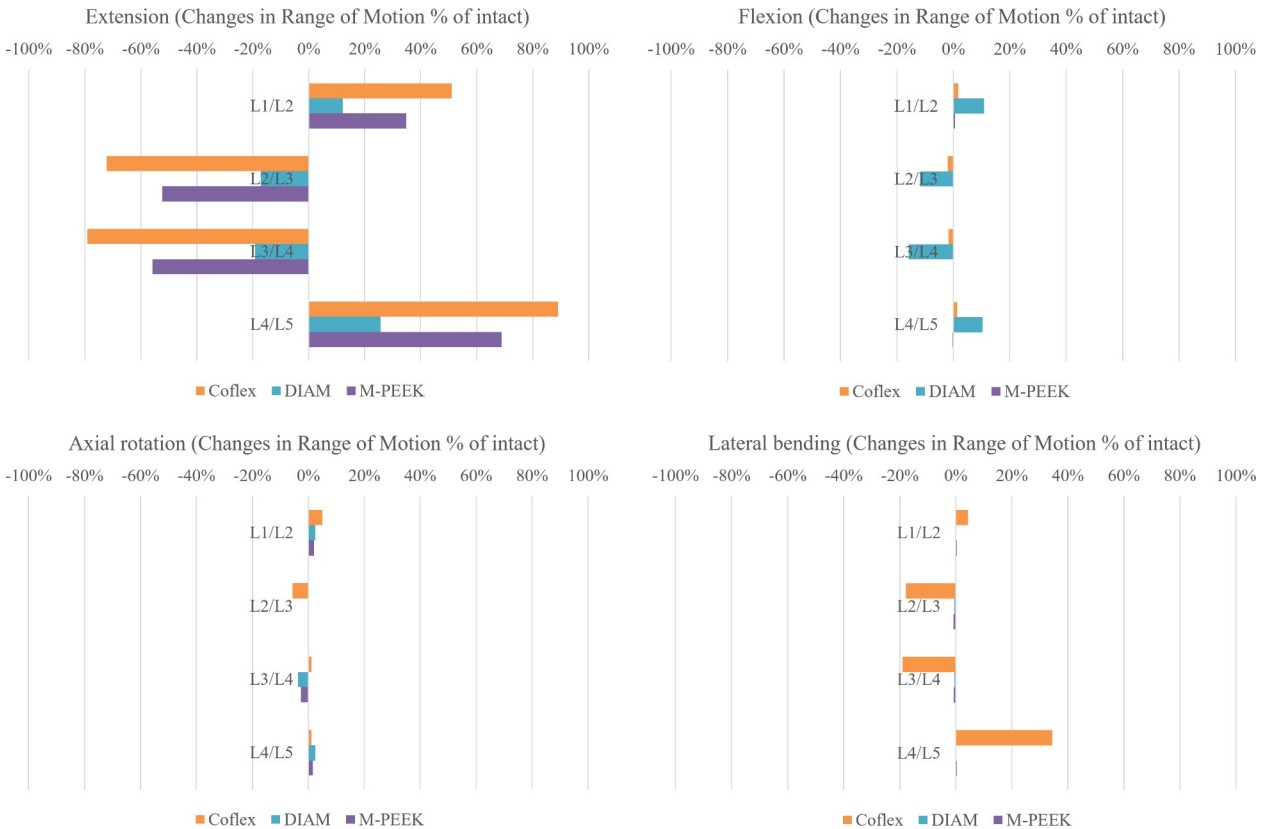

**Fig 2. Difference in ROM as a percentage of the intact model (% of intact) at the implanted and adjacent levels in flexion, extension, axial rotation, and lateral bending.**

In axial rotation, the disc stress differences between the intact, DIAM models was less than 6% for each segment; M-PEEK models was less than 2% for all segments; it was different more than 13% in all segments of Coflex model. In lateral bending, the disc stress was different to the intact less than 1% for each segment of DIAM model; it was different more than 13% in all segments of Coflex model, as expressed in Fig 3.

## Contact force on the spinous process at the instrumented level

The maximum stress on the spinous process at the instrumented levels occurred during extension. Thus, this result was focused on the Von-Mises stress on the spinous process during extension, while loading during lateral bending or rotation were not addressed. For the Coflex, DIAM and M-PEEK models, the contact forces on the L2 spinous process were 277N, 32N and 155N, respectively, and the forces on the L3 spinous process were 280N, 38N and 161N, respectively (Fig 4A). Fig 4A also shows stress concentrations at the superior and inferior surface of the L3 spinous process in the Coflex model. While the DIAM model produced the lowest stress on the inferior surface of the L2 and L3 spinous process in extension.

## Facet contact forces at the instrumented and adjacent levels under extension

Fig 4B presents the bilateral facet loads at the instrumented and adjacent levels during extension. The facet contact forces at the instrumented levels under extension were lower than those in the intact model, but were higher at the adjacent levels. Compared to the intact model in

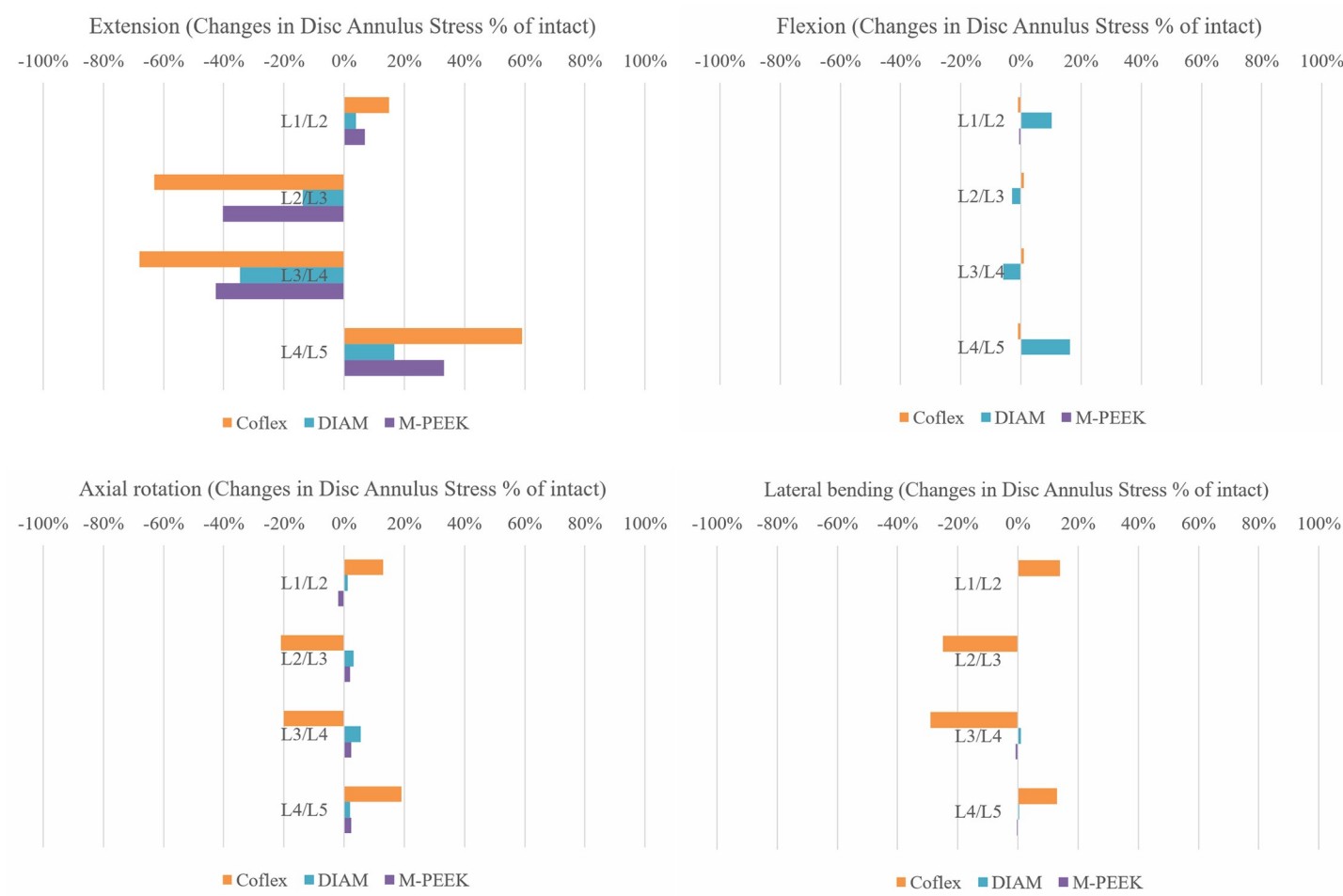

**Fig 3. Differences in annulus stress as a percentage of the intact model (% of intact) at the implanted and adjacent levels in flexion, extension, axial rotation, and lateral bending.**

extension, the contact forces decreased at the L2-3 and L3-4 instrumented level of all instrumented models. At the adjacent levels, the contact forces at L1-2 and L4-5 levels increased in all instrumented models (Fig 4B).

## Discussion

The biomechanical behavior of the lumbar spine after implanting an IPD across two levels has not been reported on until now. Thus, the goal of this work was to look into alterations in the biomechanical characteristics of the lumbar spine after implanting an IPD across two adjacent levels and then to compare the significance with a single-level instrumented model.

The previous study showed that the Coflex device acted to significantly restrain the ROM of the implanted level in extension and lateral bending, but the adjacent level was not significantly affected [23]. For the DIAM device, Bellini et al. reported that the ROM of the instrumented level decreased in both flexion and extension after DIAM insertion, whereas the ROM of the adjacent levels was unchanged [24]. It is considered that the Coflex and DIAM devices are not capable of fully compensating for an unstable spine when loaded in different directions, with the exception of extension with a single-level insertion [5, 20, 25, 26]. Similarly, the M-PEEK model constrained the ROM of instrumented levels in extension, but movements on

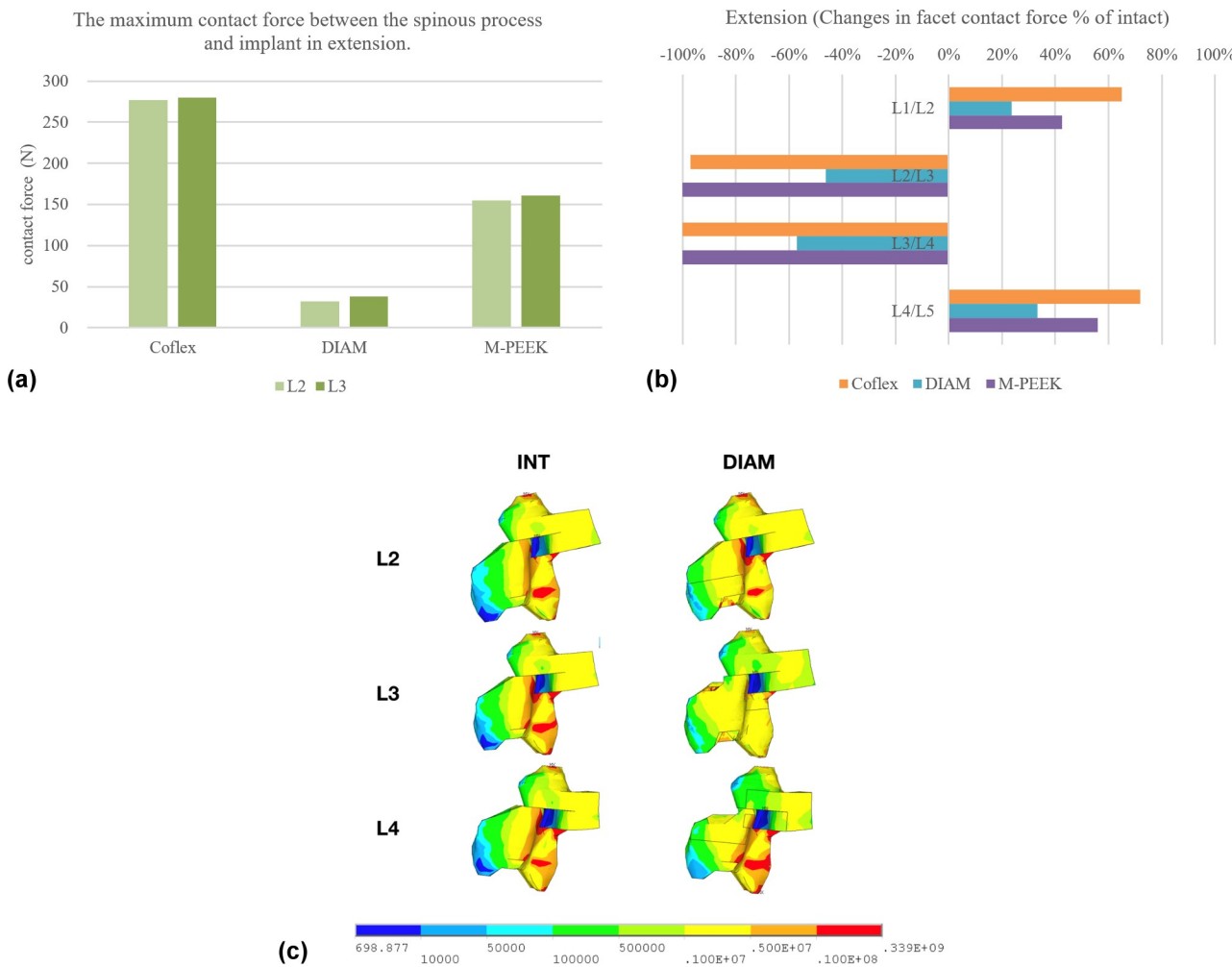

**Fig 4.** a) Maximum contact force between the spinous process in extension, b) changes in facet contact force as compared to the intact model under extension, c) stress distribution on the spinous process in extension.

other motions were not significant [15]. In this study, the changes in the ROM of the double DIAM model were smaller than Coflex and M-PEEK models at all levels in extension.

During extension, the stress on the discs in the Coflex, DIAM and M-PEEK models at the instrumented levels (L2-3 and L3-4) were significantly decreased in comparison to the baseline levels of the intact model. The Coflex model had the lowest ROM in extension at the instrumented segments and, as such, the disc stress was a lot less than in the other models. An increased ROM in extension may result in greater compressive loading on the posterior disc and increase stresses within the disc. During flexion, there was no substantial difference in disc stresses at each layer in the intact, Coflex and M-PEEK model. This is probable because the Coflex and M-PEEK devices are not designed to constrain forward flexion. The DIAM model exhibited a significant increase in stress at the L4-5 disc at the adjacent level during flexion, most likely because the DIAM device decreases motion at the implanted levels and the adjacent segments are forced to compensate for this loss of motion. During axial rotation, the DIAM and M-PEEK models did not produce any significant change in disc stress at each level, but in the Coflex model the annulus stress, increased by 13% of the adjacent L1-2 segment and 19% of the adjacent L4-5 segment. The fixation method of the Coflex device means it rigidly

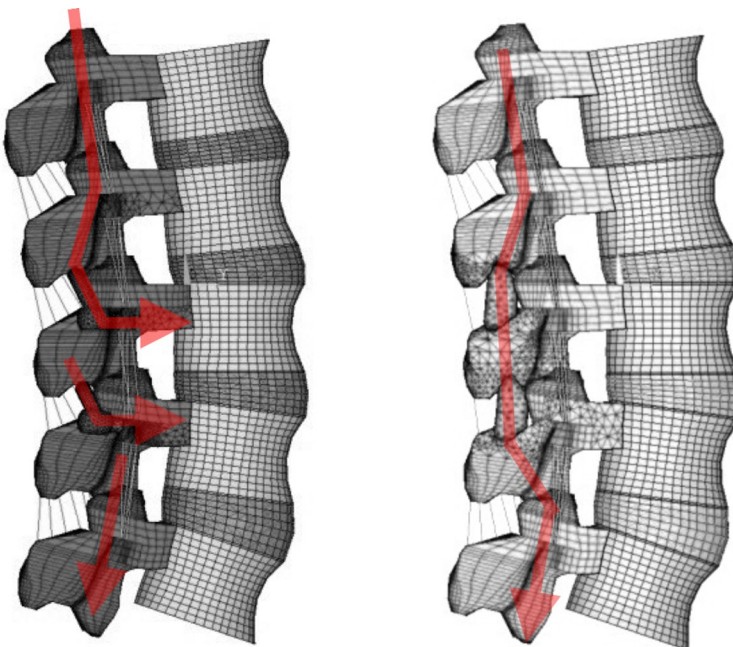

**Fig 5.** Stress path of facet joint in a) M-PEEK and b) Coflex and DIAM under extension.

holds the spinous processes at implanting segment, with the result that this type of fixation caused a higher stiffness at the implanted segments and greater stress at the adjacent levels in axial rotation. These results are in agreement with a cadaveric study from Tsai et al. [5]. In lateral bending, our results showed the Coflex model had the greatest changes in disc stress in instrumented levels of all models, with the peak stress being observed at the adjacent level. In general, these results are similar to those from studies on spinal fusion [27, 28].

Previous studies evaluating the contact forces on the facet joints reported that the highest force occurred during extension [20, 29]. As such, this study focused on the facet force during extension, while loading during lateral bending or rotation were not addressed. In all implanted models, there was a pronounced decrease in facet contract force on the instrumented segments, but both the upper and lower adjacent levels showed an increase in facet contract force, with the lower adjacent level showing the greatest increment in force. Although the Coflex and M-PEEK devices, both offered excellent resistance to extend, the M-PEEK device showed less changes in facet contact forces at adjacent levels. This is mainly down to the difference in fixation method between these devices. The M-PEEK device is fixed by two posterior pedicle screws at the implanted segment, and the associated load is transferred more towards the anterior column of the lower adjacent segment than with the Coflex or DIAM devices (Fig 5). In a FEM study of single-level implantation with a DIAM device [20], a decrease in facet forces was observed at the treated segment, but there was an increase at the adjacent segments, particularly in the lower adjacent segment. A finite element study by Byun et al. [23] revealed that the average facet contract force at the upper adjacent segment increased by 170% when using a Coflex implant, but the values from the lower adjacent segment were not recorded. A possible reason for the increase in adjacent facet force after inserting an IPD is that the instantaneous axis of rotation is shifted towards the posterior portion of the implant, leading to an increase in force being transferred through the posterior column of the adjacent facets. The results of this study showed that inserting an IPD across two levels caused a marked

increase in adjacent facet force, which may lead to an increase in the risk of adjacent facet hypertrophy.

Fracture of the spinous process is the primary complication of interspinous devices [11, 30, 31]. A cadaveric study by Shepherd et al. [32] recorded the average failure load of an intact spinous process as 339 N under a superiorly-directed load from a spinous process device. In this study, the maximum contact force on the spinous processes occurred in the L3 process in all instrumented models and is safe within the strength of the bone. Nevertheless, it should also be considered that under repeated loading, the load to failure may decrease and cause a fracture of the spinous process. This is one of the limitations of this study.

Likewise, the high contact force at the spinous processes of instrumented levels recorded in the Coflex model demonstrated the risk of spinous process fracture. The sandwich phenomenon of the intervening spinous process in patients with adjacent, double-level IPDs has been demonstrated [11]. The phenomenon may lead to fracture of the spinous process between the double IPDs and subsequent implant migration or failure. In contrast, the sandwich phenomenon is unlikely to occur with the use of the M-PEEK or DIAM device. The M-PEEK device creates a load sharing from the superior spinous process to the lower pedicle screws, instead of to the lower spinous process, and the lower stiffness of the DIAM device offers superior load-sharing capacity between the two bridge supports (Fig 4C).

There are several limitations of this study that arise from the simplified finite element models. All vertebral bodies were simplified to the same shape, but the size was scaled to match each individual vertebral dimension from the radiography images. The material properties of all vertebrae bodies were also assumed as homogenous and isotropic, which might not be true for each vertebra. The simulation revealed that the ROM of instrumented levels in Coflex model in flexion was higher than DIAM model, this might be due to removing the supraspinous and interspinous ligaments. But literatures revealed that lumbar intervertebral disc and facet joints are the major load carrier during the functional unit motion. Further, the Young moduli of the supraspinous and interspinous ligaments in this study were 8 and 10Mpa, respectively, that are much smaller value in comparison with the cortical bone and implants. Therefore, we hypothesized that both ligaments might have a limited contribution in ROM, the main cause of larger instrumented level motion is mainly due to the dimension and location of the Coflex implant as the pivot of extension. Moreover, the loading conditions were not precisely the same as a physiological loading because these finite element models could not simulate as a real muscle contraction. As well, the adjacent segments must compensate more for ROM when using rigid implants than mobile devices [33], because the use of the hybrid method [21]. The ligamentous lumbar spine model performs a low resistance initially, while past its neutral position the ligaments mechanical stiffness increased gradually when as load increased. Therefore, the ROM of adjacent segments may vary [33] even all segments are subject to the same loading.

## Conclusions

Of the three devices, the Coflex implant shows the limited ROM at the instrumented level but compensating larger ROM at the two adject segments. In combined with highest annulus stress on adjacent level, which may accelerate the degeneration of the adjacent segments. This might be the factor of the sandwich phenomenon at the spinous process was only happened in Coflex model. The DIAM and M-PEEK devices offered superior load sharing and could be expected to be at lower risk of developing adjacent level degeneration as well as spinous process fracture. The design features such as load transmission from the superior spinous process

to lower vertebra and offering superior load-sharing capacity between the two bridge supports, should be considered in future implant designs.

## Author Contributions

**Conceptualization:** Hung-Ming Chen, Yi-Jie Kuo, Sai-Wei Yang.

**Software:** Hao-Ju Lo.

**Validation:** Hao-Ju Lo.

**Writing – original draft:** Hao-Ju Lo.

**Writing – review & editing:** Hung-Ming Chen, Yi-Jie Kuo, Sai-Wei Yang.

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
