## [Decision Letter · Decision Letter 0]

29 Jun 2020

PONE-D-20-06911

Effect of different designs of interspinous process devices on the instrumented and adjacent levels after double-level lumbar decompression surgery: A finite element analysis

PLOS ONE

Dear Dr. Yang,

Thank you for submitting your manuscript to PLOS ONE. After careful consideration, we feel that it has merit but does not fully meet PLOS ONE’s publication criteria as it currently stands. Therefore, we invite you to submit a revised version of the manuscript that addresses the points raised during the review process.

We look forward to receiving your revised manuscript.

Kind regards,

Osama Farouk

Academic Editor

PLOS ONE

Journal Requirements:

Additional Editor Comments (if provided):

Reviewers' comments:

Reviewer's Responses to Questions

**Comments to the Author**

1. Is the manuscript technically sound, and do the data support the conclusions?

Reviewer #1: Yes

Reviewer #2: Yes

2. Has the statistical analysis been performed appropriately and rigorously? 

Reviewer #1: Yes

Reviewer #2: I Don't Know

3. Have the authors made all data underlying the findings in their manuscript fully available?

Reviewer #1: No

Reviewer #2: Yes

4. Is the manuscript presented in an intelligible fashion and written in standard English?

Reviewer #1: Yes

Reviewer #2: Yes

5. Review Comments to the Author

Reviewer #1: The authors present a finite element study on 3 interspinous devices and analyse the range of motion, stress on the disc and facet joints as well as contact forces on the spinous processes to investigate the causes of spinal process fractures.

The authors could find that the devices have different Impacts on the spinal structures, especially in 2-segment Implantation.

I have several issues:

- The M-PEEK was developed in the institution of the authors. Please state the connection to the developer of the device.

- The function and validity of the ANSYS Software should be explained in more detail (at least 2-3 sentences), please do not refer to other studies only.

Reviewer #2: General comments

. Meaning of any abbreviation must be mentioned on its first use in the abstract (as a separate entity) and in the manuscript text. Afterword, the abbreviation must be used instead of the original word. The term interspinous devices was used throughout the introduction and then the abbreviation IPD was used from line 80. Lumbar spinal stenosis was abbreviated as (LSS) in line 40 and then the full term (lumbar spinal stenosis) was used in the "Discussion" section in lines 217-218.

For any term used one time, no need to be abbreviated e.g Intermittent neurogenic claudication (INC) (line 48)

.Figures, figure legend and tables must be moved after the end of the text. Then figures and tables are cited in the text. Examples: lines 120-122: “Fig 1 Finite element model of the a) intact lumbar spine implanted with two interspinous spacers of (b) DIAM, (c) pedicle screw-based M-rod (M-PEEK) system and (d) Coflex at L2/L3 and L3/L4 segments”. Lines 172-173 & “Fig 2. Difference in ROM as a percentage of the intact model (% of intact) at the

implanted and adjacent levels in flexion, extension, axial rotation, and lateral bending” must be moved to Figure legend after the references

Abstract

Lines 22-23; "Recently, various designs and material manufactured interspinous process devices (IPDs)

23 are on the market in managing symptomatic lumbar spinal pathology" Lumbar spinal pathology should be changed to "Lumbar spinal stenosis".

Introduction

Is too long with a lot of repetitions. It must be shortened and concentrate on the aim of the current study.

Line 85: “..middle spinous process…” what does it mean? Middle in relation to what?

Results

The authors did not perform any statistical testing for their results

. Lines 177-178: “..for the Coflex model, the ROM was more variation as compared to the intact

model, as depicted in Fig 2 and Table 2.”

What is the meaning of this word?

. Discussion

is long and should be shortened

. Line 237: “Effect“ is better to be used than “result”

. Lines 316-317: “Also, the adjacent segments must compensate more for range of motion when using rigid

implants than mobile devices [33] because the use of the hybrid method.” What is the meaning of this word?

6. PLOS authors have the option to publish the peer review history of their article (what does this mean?). If published, this will include your full peer review and any attached files.

Reviewer #1: No

Reviewer #2: Yes: Mohamed E Abdel-Wanis

---

## [Author Response · Author response to Decision Letter 0]

3 Sep 2020

Response to reviewers:

The modified lines mentioned below are indicated in the text by highlighting in the file "Revised Manuscript with hightlight for response.docx"

Reviewer #1: The authors present a finite element study on 3 interspinous devices and analyse the range of motion, stress on the disc and facet joints as well as contact forces on the spinous processes to investigate the causes of spinal process fractures.

The authors could find that the devices have different Impacts on the spinal structures, especially in 2-segment Implantation. 

I have several issues:

- The M-PEEK was developed in the institution of the authors. Please state the connection to the developer of the device.

Response: The first author, corresponding author and the developer (Dr. Hsin-Chang Chen) of M-PEEK device were colleagues in Department of Biomedical Engineering, National Yang-Ming University, Taipei, Taiwan. We study the same field of spinal biomechanics.

- The function and validity of the ANSYS Software should be explained in more detail (at least 2-3 sentences), please do not refer to other studies only. 

Response: The statements of the model validation have been added in Lines 100-105 as follows, “The intact model was validated by applying a pure moment of 7.5 Nm in flexion, extension, torsion, and lateral bending, respectively; all degrees of freedom at the inferior surfaces of the L5 vertebra were constrained. The ROM of each vertebra and the facet joint forces at all segments was computed. Subsequently, 1000 N axial compressive force was acting on the top of L1 vertebra and the intradiscal pressures were calculated in comparing to published literatures [16-18].” 

Reviewer #2: General comments

. Meaning of any abbreviation must be mentioned on its first use in the abstract (as a separate entity) and in the manuscript text. Afterword, the abbreviation must be used instead of the original word. The term interspinous devices was used throughout the introduction and then the abbreviation IPD was used from line 80. Lumbar spinal stenosis was abbreviated as (LSS) in line 40 and then the full term (lumbar spinal stenosis) was used in the "Discussion" section in lines 217-218.

For any term used one time, no need to be abbreviated e.g Intermittent neurogenic claudication (INC) (line 48) 

Response: The abbreviation problems of IPD, LSS and INC have been fixed in revised manuscript.

.Figures, figure legend and tables must be moved after the end of the text. Then figures and tables are cited in the text. Examples: lines 120-122: “Fig 1 Finite element model of the a) intact lumbar spine implanted with two interspinous spacers of (b) DIAM, (c) pedicle screw-based M-rod (M-PEEK) system and (d) Coflex at L2/L3 and L3/L4 segments”. Lines 172-173 & “Fig 2. Difference in ROM as a percentage of the intact model (% of intact) at the implanted and adjacent levels in flexion, extension, axial rotation, and lateral bending” must be moved to Figure legend after the references

Response: The figure legend and tables have been moved after the end of the text in the revised manuscript.

Abstract

Lines 22-23; "Recently, various designs and material manufactured interspinous process devices (IPDs) 23 are on the market in managing symptomatic lumbar spinal pathology" Lumbar spinal pathology should be changed to "Lumbar spinal stenosis".

Response: The term “Lumbar spinal pathology” has been changed to “Lumbar spinal stenosis” in Line 23 in our revised manuscript.

Introduction

Is too long with a lot of repetitions. It must be shortened and concentrate on the aim of the current study.

Response: Thank you for review’s suggestion. The introduction section has been shortened and concentrate on the aim of the current study as our revised manuscript.

Line 85: “..middle spinous process...” what does it mean? Middle in relation to what?

Response: The “..middle spinous process...” means “the intervening spinous process in patients with adjacent, double-level IPDs”. This term has been rewriting as “the intervening spinous process in patients with adjacent, double-level IPDs” in Line 80 and Line 251.

Results

The authors did not perform any statistical testing for their results

Response: The results of ROM or contact force in the specific segmentation or disc of specific model in this study is unique and repeatable when the boundary condition, loading condition and material properties were the same. Therefore, we did not perform any statistical testing for our results.

. Lines 177-178: “..for the Coflex model, the ROM was more variation as compared to the intact model, as depicted in Fig 2 and Table 2.” What is the meaning of this word?

Response: This sentence means the ROM differences between the Coflex and INT models are larger and obviously in lateral bending. In contract, the ROM differences between the DIMA/M-PEEK and INT models are very similar.

. Discussion is long and should be shortened

Response: Thank you for review’s suggestion. The discussion section has been shortened and concentrate on the aim of the current study as our revised manuscript.

. Line 237: “Effect“ is better to be used than “result”

Response: The “but the result on other movements was not significant” has been modified as “but movements on other motions were not significant” in Line 198-199 of our revised manuscript.

. Lines 316-317: “Also, the adjacent segments must compensate more for range of motion when using rigid implants than mobile devices [33] because the use of the hybrid method.” What is the meaning of this word? 

Response: With the use of the hybrid method, the moment placed on the fusion segment increases proportionally to the additional adjacent segment motion. Therefore, adjacent segments must compensate more when using rigid implants than mobile devices [33]. The stiffness of the adjacent segments directly impacts the motion distribution among these segments. Due to its nonlinear behavior, the spine offers low resistance to movement when in its neutral position, but gradually stiffens when loaded. This means that the stiff adjacent segments will typically have a lower range of motion than mobile segments. Therefore, even though all segments are subjected to the same loading, the mobility of adjacent segments may vary.

---

## [Decision Letter · Decision Letter 1]

23 Sep 2020

PONE-D-20-06911R1

Effect of different designs of interspinous process devices on the instrumented and adjacent levels after double-level lumbar decompression surgery: A finite element analysis

PLOS ONE

Dear Dr. Yang,

Thank you for submitting your manuscript to PLOS ONE. After careful consideration, we feel that it has merit but does not fully meet PLOS ONE’s publication criteria as it currently stands. Therefore, we invite you to submit a revised version of the manuscript that addresses the points raised during the review process.

The device developer and the authors seem to work in the same Institution and work closely together. Conflict of interest statement is required.

We look forward to receiving your revised manuscript.

Kind regards,

Osama Farouk

Academic Editor

PLOS ONE

Reviewers' comments:

Reviewer's Responses to Questions

**Comments to the Author**

1. If the authors have adequately addressed your comments raised in a previous round of review and you feel that this manuscript is now acceptable for publication, you may indicate that here to bypass the “Comments to the Author” section, enter your conflict of interest statement in the “Confidential to Editor” section, and submit your "Accept" recommendation.

Reviewer #1: All comments have been addressed

Reviewer #2: (No Response)

2. Is the manuscript technically sound, and do the data support the conclusions?

Reviewer #1: Yes

Reviewer #2: Yes

3. Has the statistical analysis been performed appropriately and rigorously? 

Reviewer #1: Yes

Reviewer #2: N/A

4. Have the authors made all data underlying the findings in their manuscript fully available?

Reviewer #1: No

Reviewer #2: Yes

5. Is the manuscript presented in an intelligible fashion and written in standard English?

Reviewer #1: Yes

Reviewer #2: No

6. Review Comments to the Author

Reviewer #1: The authors have addressed all comments adequately.

The authors have addressed all comments adequately.

Reviewer #2: General comments

English language is poor and must be improved

Introduction

Lines 51-52: “The lumbar interspinous process device (IPD) is based on this principle and is placed at

a symptomatic level to limit the extension of the lumbar spine” What do the authors mean by “this principle”?

Repetitions are still present: authors reported mechanism of action of these IPDs at 3 sites in the Introduction only:

. Lines 51-53: …and is placed at

a symptomatic level to limit the extension of the lumbar spine and maintain a relatively flexible

mobility at the intradiscal level to achieve the goal of symptom relief.

. Lines 57-58; These IPDs are designed to enlarge the canal space and to relieve pain by offloading the

entrapped spinal root nerves and as the intralaminar stabilizer .

. Lines 59-62: Such devices provide significant

improvements in managing the spinal pathology by rapid indirect decompression, and

progressively decrease intradiscal and facet loads, which restores the neuroforaminal spaces

and stabilize the spinal column in different postures

. Lines 71-72: What do the authors mean by “with limited disclosure information” ?

. Line 87: This study aimed to investigate the effect of different designs and material used

of the IPDs in the management of neurological pain due to LSS”

This study did not investigate any point related to effect of these IPD on neurological pain

. The goal of the syudy was not clarified in the “Introduction” but clarified in the “Discussion” Lines 187-190 to be “The goal of this work was to look into

alterations in the biomechanical characteristics of the lumbar spine after implanting an IPD across two adjacent levels and then to compare the significance with a single-levelinstrumented model.” I would recommend that the authors clarify the aim of the work in the “Introduction”

.

. Material and Methods

. Lines 188-119: the supraspinous and interspinous ligaments were

preserved in both the DIAM and M-PEEK models. Do this make difference between the models that may affect the results?

The authors must answer this question in the discussion

Conclusions

. “Of the three devices, the Coflex implant resulted in the most limited ROM in extension

and highest annulus stress on adjacent level, which may result in accelerated degeneration of

the adjacent segments as the sandwich phenomenon”.

. What is the relation between the accelerated degenerative process and the fracture of the spinous process of the middle vertebra on applying IPD at 2 adjacent level?

7. PLOS authors have the option to publish the peer review history of their article (what does this mean?). If published, this will include your full peer review and any attached files.

Reviewer #1: No

Reviewer #2: **Yes: **Prof Mohamed E Abdel-Wanis

---

## [Author Response · Author response to Decision Letter 1]

7 Nov 2020

Dear Editor and reviewers:

Thank you so much for your work in reviewing this manuscript and your comments are addressed below. 

Response to reviewers:

Reviewer #1: The authors have addressed all comments adequately.

Thanks very much for your work in review this manuscript and we have made the changes/corrections you have suggested.

Reviewer #2: General comments

English language is poor and must be improved.

Thank you for these comments. It is good to learn when intended messages do not translate well to the reader due to the English written. we have had a professional scientific editor to proofread and polish this manuscript and adjusted the section on clustering

Introduction

Lines 51-52: “The lumbar interspinous process device (IPD) is based on this principle and is placed at a symptomatic level to limit the extension of the lumbar spine” What do the authors mean by “this principle”? 

The “this principle” implies the “decompressive laminectomy with the dorsal decompression procedure” this is the common surgical principle, which addressed in the first paragraph of Introduction. We added “dorsal decompression” to more clarification, at Line 50 (clean MS)

Repetitions are still present: authors reported mechanism of action of these IPDs at 3 sites in the Introduction only:

Lines 51-53: …and is placed at a symptomatic level to limit the extension of the lumbar spine and maintain a relatively flexible mobility at the intradiscal level to achieve the goal of symptom relief.

Lines 57-58; These IPDs are designed to enlarge the canal space and to relieve pain by offloading the entrapped spinal root nerves and as the intralaminar stabilizer.

Lines 59-62: Such devices provide significant improvements in managing the spinal pathology by rapid indirect decompression, and progressively decrease intradiscal and facet loads, which restores the neuroforaminal spaces and stabilize the spinal column in different postures.

Thanks your comment, the three paragraphs were attempted to address the mechanism of IPD, it did sound redundant we have rewritten this paragraph and coincide the sentences as shown at Line 50 to 56 

Lines 71-72: What do the authors mean by “with limited disclosure information” ?

The FDA approved the surgical procedure limited to two segments only, however, some surgeons might have over two-level implication without revealing in available literatures. We have deleted the words to avoid the confuse adding. 

Line 87: This study aimed to investigate the effect of different designs and material used of the IPDs in the management of neurological pain due to LSS” This study did not investigate any point related to effect of these IPD on neurological pain. The goal of the study was not clarified in the “Introduction” but clarified in the “Discussion” Lines 187-190 to be “The goal of this work was to look into alterations in the biomechanical characteristics of the lumbar spine after implanting an IPD across two adjacent levels and then to compare the significance with a single level instrumented model.” I would recommend that the authors clarify the aim of the work in the “Introduction”. 

Thanks, your valued suggestion, the aim of the work had been moved from Discussion section and clarified in the Introduction section in our revised manuscript. In addition, the effect of IPDs on neurological pain was also removed from the goal of this work.

Material and Methods

Lines 188-119: the supraspinous and interspinous ligaments were preserved in both the DIAM and M-PEEK models. Do this make difference between the models that may affect the results? The authors must answer this question in the discussion.

As indicated at Line 107, due to design consideration and following real surgical procedures, in the Coflex surgical procedure, the supraspinous and interspinous ligaments have to dissect in order to insert the implant. 

As far as the contribution of the motion restrain of the ISL and SSL, the Young modulus in this study are 10 and 8 Mpa for ISL and SSL, respectively. Which is very small value in comparison with the cortical bone and implants. Literatures revealed that lumbar intervertebral disc and facet joints are the major load carrier during the functional unit motion. Our results revealed that this difference between Coflex and DIAM/M-PEEK affected the ROM in flexion. The ROM of instrumented levels in Coflex model in flexion was higher than DIAM model, removing the supraspinous and interspinous ligaments might be one of the factors, but the dimension and location of the implant as the pivot of extension could be the major cause. Nevertheless, it did not affect our main findings of adjacent level problems and a fracture of the spinous process. In addition, this is also a limitation in this study to have identical ligaments constraints in each model. Thank you for the comment and your inquiry was addressed in the discussion of this study, starting at the Line 257-265.

Conclusions

“Of the three devices, the Coflex implant resulted in the most limited ROM in extension and highest annulus stress on adjacent level, which may result in accelerated degeneration of the adjacent segments as the sandwich phenomenon”. What is the relation between the accelerated degenerative process and the fracture of the spinous process of the middle vertebra on applying IPD at 2 adjacent level?

So far there is no study reveals the relationship between degeneration of the adjacent segments and the sandwich phenomenon on spinous process. This statement had been rewritten as “Of the three devices, the Coflex implant shows the limited ROM at the instrumented level but compensating larger ROM at the two adject segments. In combined with highest annulus stress on adjacent level, which may accelerate the degeneration of the adjacent segments.” in Conclusion section in our revised manuscript. Staring line 275-277

Thank you again for all your comments. They have helped us develop the manuscript considerably.

---

## [Decision Letter · Decision Letter 2]

14 Dec 2020

Effect of different designs of interspinous process devices on the instrumented and adjacent levels after double-level lumbar decompression surgery: A finite element analysis

PONE-D-20-06911R2

Dear Dr. Yang,

We’re pleased to inform you that your manuscript has been judged scientifically suitable for publication and will be formally accepted for publication once it meets all outstanding technical requirements.

Kind regards,

Osama Farouk

Academic Editor

PLOS ONE

Additional Editor Comments (optional):

Reviewers' comments:

Reviewer's Responses to Questions

**Comments to the Author**

1. If the authors have adequately addressed your comments raised in a previous round of review and you feel that this manuscript is now acceptable for publication, you may indicate that here to bypass the “Comments to the Author” section, enter your conflict of interest statement in the “Confidential to Editor” section, and submit your "Accept" recommendation.

Reviewer #1: All comments have been addressed

Reviewer #2: All comments have been addressed

2. Is the manuscript technically sound, and do the data support the conclusions?

Reviewer #1: Yes

Reviewer #2: Yes

3. Has the statistical analysis been performed appropriately and rigorously? 

Reviewer #1: Yes

Reviewer #2: N/A

4. Have the authors made all data underlying the findings in their manuscript fully available?

Reviewer #1: Yes

Reviewer #2: Yes

5. Is the manuscript presented in an intelligible fashion and written in standard English?

Reviewer #1: Yes

Reviewer #2: Yes

6. Review Comments to the Author

Reviewer #1: The authors present a finite elemnt analysis on the effect of different interspinous process devices on the instrumented and adjacent levels and provided adequate comments to all issues.

Reviewer #2: (No Response)

7. PLOS authors have the option to publish the peer review history of their article (what does this mean?). If published, this will include your full peer review and any attached files.

Reviewer #1: **Yes: **PD Dr. Mario Cabraja

Reviewer #2: **Yes: **Mohamed Abdel-Wanis

---

## [Editor Report · Acceptance letter]

18 Dec 2020

PONE-D-20-06911R2 

Effect of different designs of interspinous process devices on the instrumented and adjacent levels after double-level lumbar decompression surgery: A finite element analysis 

Dear Dr. Yang:

I'm pleased to inform you that your manuscript has been deemed suitable for publication in PLOS ONE. Congratulations! Your manuscript is now with our production department. 

Kind regards, 

on behalf of

Dr. Osama Farouk 

Academic Editor

PLOS ONE